



# Evaluation of INCA precipitation analysis using a very dense rain gauge network in southeast Austria

Esmail Ghaemi[1, 2, 3], Ulrich Foelsche[1, 2, 3], Alexander Kann[4], Jürgen Fuchsberger[3]

[1]Institute for Geophysics, Astrophysics, and Meteorology/Institute of Physics (IGAM/IP), NAWI Graz, University of Graz, Austria
[2]FWF-DK Climate Change, University of Graz, Austria
[3]Wegener Center for Climate and Global Change (WEGC), University of Graz, Austria
[4]Department of Forecasting Models, Central Institute for Meteorology and Geodynamics (ZAMG), Vienna, Austria

*Correspondence to*: Esmail Ghaemi (esmail.ghaemi@uni-graz.at)

**Abstract.** An accurate estimate of precipitation is essential to improve the reliability of hydrological models and helps for decision-making in agriculture and economy. Merged radar–rain-gauge products provide precipitation estimates at high spatial and temporal resolution. In this study, we assess the ability of the INCA (Integrated Nowcasting through Comprehensive Analysis) precipitation analysis product provided by ZAMG (the Austrian Central Institute for Meteorology and Geodynamics) in detecting and estimating precipitation for 12 years in southeast Austria. The blended radar–rain-gauge INCA precipitation analyses are evaluated using WegenerNet – a very dense rain gauge network with about 1 station per 2 km$^2$ – as 'true precipitation'. We analyze annual, seasonal, and extreme precipitation of the 1 km × 1 km INCA product and its development from 2007 to 2018. Based on the results, the performance of INCA can be divided into three different periods. From 2007 to 2011, the annual area-mean precipitation in INCA was slightly higher than WegenerNet, except in 2009. However, INCA underestimates precipitation in grid cells farther away from the two ZAMG meteorological stations in the study area (which are used as input for INCA), especially from May to September ("wet season"). From 2012 to 2014, INCA's overestimation of the annual-mean precipitation amount is even higher, with an average of 25 %, but INCA performs better close to the two ZAMG stations. From 2015 onwards, the overestimation is still dominant in most cells but less pronounced than during the second period, with an average of 12.5 %. Regarding precipitation detection, INCA performs better during the wet seasons. Generally, false events in INCA happen less frequently in the cells closer to the ZAMG stations than in other cells. The number of true events, however, is comparably low closer to the ZAMG stations. The difference between INCA and WegenerNet estimates is more noticeable for extremes. We separate individual events using a 1-hour minimum inter-event time (MIT) and demonstrate that INCA underestimates the events' peak intensity until 2012 and overestimates this value after mid-2012 in most cases. The overestimation of the peak-intensity is more pronounced during July. In general, the precipitation rate and the number of grid cells with precipitation are higher in INCA. Furthermore, 40 % of the individual events start earlier, and 50 % end later in INCA. Considering four extreme convective short-duration events, there is a time shift in peak intensity detection. The relative differences in the peak intensity in these events can



change from approximately -40 % to 40 %. The results of this study can be used for further improvements of INCA products as well as for future hydrological studies in this area.

## 1 Introduction

Precipitation is one of the most important components of the hydrological cycle and plays a crucial role in shaping the Earth's climate. It can trigger numerous natural hazards such as flash floods, soil erosion, and landslide, which often jeopardize human life and can cause tremendous economic loss. More accurate precipitation estimates improve the reliability of hydrological models and numerical weather prediction models (NWPs), and can lead to a better understanding of uncertainties in climate model outputs. Furthermore, having a reliable estimate of precipitation is vital for decision making in hydrology, agriculture, and economy. Since the characteristics of a precipitation event can change rapidly in both time and space, accurate estimates with high spatial and temporal resolution remain challenging, especially in smaller scale events.

Rain gauges have been used as direct measuring devices to estimate precipitation for decades. Besides station-based data, remote sensing estimates, such as weather radar and satellite data, are meanwhile widely used. Each of these approaches has its strengths and weaknesses. For instance, rain gauges are more accurate in measuring intensity due to their direct measurement techniques. However, a highly dense gauge network is required to detect small-scale convective events. Moreover, gauge data are subject to different types of errors, such as wind-undercatch (Habib et al., 2001; Pollock et al., 2018). Furthermore, most rain gauges in mountainous areas are located in the valleys, which can lead to an underestimation of orographic rainfall in those areas (Ebert et al., 2007).

On the other hand, satellites can cover the entire globe and weather radars have a high spatial resolution. They can estimate precipitation using various ranges of electromagnetic waves. Radar estimates are based on converting reflectivity of hydrometeors to rain rate (also known as the Z-R relationship), and different sources of errors and uncertainties can have considerable effects on these estimates. Beam over-shooting, partial beam filling, non-uniformity in the vertical profile of reflectivity (VPR), hardware calibration, a fixed Z-R relationship for different precipitation types, and random sampling errors are some examples of weather radar errors (AghaKouchak, 2010).

In general, considering these two approaches as complementary and merging them can lead to more reliable estimates with a higher resolution (Goudenhoofdt and Delobbe, 2009). Using multiple sources of data, including radar, gauges, and model outputs is beneficial to overcome some of the limitations addressed above (Ayat et al., 2021). However, the aforementioned errors and weaknesses in both rain gauges and radar estimates can still affect the reliability of the merged data and need to be considered (Haiden et al., 2011). The Integrated Nowcasting through Comprehensive Analysis (INCA) of the Austrian Central Institute for Meteorology and Geodynamics (ZAMG) provides high-resolution precipitation analyses and nowcasts by combining ground station, remote sensing, high-resolution topographic data, and NWP data. INCA's meteorological products are used, for example, as inputs for flood forecasting in the Alpine region and winter rail maintenance (Kann and Haiden, 2011).





The aim of this study is to evaluate INCA precipitation analyses over a period of 12 years, using gridded precipitation fields from the dense WegenerNet weather and climate station network in southeast Austria. The main focus lies on analyzing the ability of INCA to detect and estimate precipitation, and on studying the impact of modifications of INCA algorithms and input data during these 12 years. We analyze annual data, seasonal data, and extremes, using different metrics. Moreover, INCA's detection skill is studied using categorical metrics. Furthermore, we identify individual events using a simple

threshold based on the interval between two consecutive events and compare the events' characteristics in both datasets. Finally, we separately study extreme convective short-duration events and demonstrate four representative examples. The following research questions are addressed and discussed in this study:

1.      How well can INCA detect and estimate precipitation in an area with a moderate topography?

2.      How did the developments in the Austrian radar network affect INCA's performance?

3.      How reliable are INCA estimates of extremes?

This paper is structured as follows. In Sect. 2, we introduce the study area and each dataset's main features; in Sect. 3, the methodology is described. The results based on different time scales and individual events are discussed in Sect. 4, and we conclude in Sect. 5.

## 2 Study area and datasets

### 2.1 WegenerNet

The WegenerNet network is a dense climate station network located in the Feldbach region in southeast Austria (see Fig. 1). The network includes 155 ground stations, almost uniformly spread over an area of about 22 km × 16 km (i.e., about one station per 2 km$^2$) provided by the Wegener Centre for Climate and Global Change, University of Graz, Austria (Kirchengast et al., 2014; Fuchsberger et al., 2020b). The highest altitude in this region is 609 m above Mean Sea Level (MSL), located in

the Southern part. The altitude decreases northward to the valley of the river Raab (see Fig. 1). The Feldbach region is affected by both Mediterranean and continental climates. Most of the precipitation occurs from May until September (here defined as the "wet season"), when monthly average precipitation is approximately twice as high as during the "dry season" from October to April (O and Foelsche, 2019). Considering that the average number of days with fresh snow in this area is less than 15 days during 1971 to 2000 (Prettenthaler et al., 2010), and has been decreasing over time, snowfall is relatively

unimportant in this area.


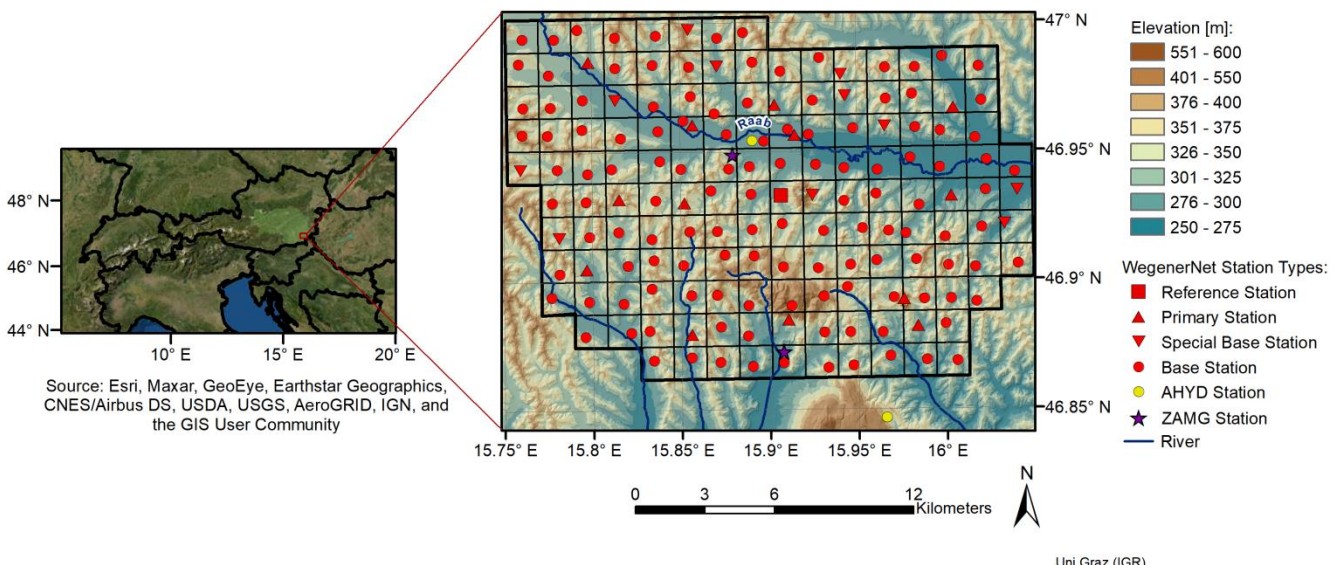

**Figure 1.** Location and topography of the study area and location of WegenerNet stations.

The WegenerNet provides station time series and gridded datasets of temperature, humidity, precipitation, and other parameters with 5 min temporal resolution since January 2007. The raw measurements are checked by seven layers of the

Quality Control System (QCS), from checking station operation and availability to interstation consistency. For detailed descriptions of the quality control system, we refer to Kabas et al. (2011), Kirchengast et al. (2014), Scheidl (2014), and Fuchsberger et al. (2020b). In this study, we used WegenerNet gridded data from WegenerNet's level 2 data (Fuchsberger et al., 2020a), generated with the inverse-distance-squared weighting method based on quality-controlled station data, and provided on a 200 m grid. WegenerNet data have been validated against data from operational weather stations (O et al.,

2018) and is shown to have a reliable performance in terms of magnitude, frequency, and the exact location of extreme events (O and Foelsche, 2019). The data have been used as a reference in multiple validation studies, a selection of which is addressed below.

Kann et al. (2015) used WegenerNet data to validate 6 months of INCA data (see Sect. 2.2) and O et al. (2017) used WegenerNet data as a reference to evaluate satellite data from the Global Precipitation Measurements (GPM) mission. Based

on half-hourly Integrated Multi-satellite Retrievals for GPM (IMERG) rainfall estimates for the period of April–October in 2014 and 2015, the results indicate that all IMERG products perform better in estimating moderate rainfall (0.3 to 3 mm per 30 min) than light and heavy rainfall events. In general, IMERG Early and IMERG Late overestimate low rain rates and all three IMERG runs tend to underestimate heavy rain. In another study, Lasser et al. (2019) directly evaluated space-based GPM Dual-frequency Precipitation Radar (DPR) estimates using WegenerNet gridded fields and showed that GPD DPR

products tend to underestimate rainfall.



O and Foelsche (2019) analyzed the spatial variability of heavy rainfall events using WegenerNet gridded data. In addition, they described the dependency of area-mean rainfall on the number of gauges and temporal resolution during heavy events. The study showed that from May to September, the spatial variability in rainfall is higher than from October to April, due to a higher proportion of convective events. Based on their results, the high density and the regular distribution of WegenerNet stations generate spatially homogeneous gridded rainfall fields. A complete up-to-date list of WegenerNet-related literature can be found at https://wegcenter.uni-graz.at/en/wegenernet/publications/ (last access: 16 January 2021).

## 2.2 INCA

The INCA precipitation analysis provides data on a 1 km × 1 km spatial grid with 15 min temporal resolution, using a combination of rain gauge data, weather radar estimates, and high-resolution topography (Haiden et al., 2011). The following data are used as input for generating the INCA precipitation analysis product:

- The topography, based on digital elevation data provided by the United States Geological Survey (USGS).
- Precipitation data from about 250 semi-automatic ground stations (Teilautomatische Wetterstationen, TAWES), operated by the ZAMG, with an average interstation distance of 18 km, all of them located in Austria and two of them in the study area. Note that more stations have been added to the INCA analysis algorithm during the study period.
- Precipitation data from the Austrian hydrographic service (AHYD), which were added over time.
- Radar data from five Austrian C-band radars, supplemented by data from weather radars of neighboring countries. Starting from 2011, four of the Austrian radars were replaced by new ones (see Table B1 in Appendix B for more details).

INCA data are generated based on a Lambert conformal conic projection as a coordinate system, with reference latitudes 46° N and 49° N, and a central reference point at 47°30' N, 13°20' E.

Steps taken to produce INCA precipitation analyses are described by Haiden et al. (2011). A brief summary of this process is given below:

1. Interpolation of station data using inverse-distance-squared weighting. Note that there are two ZAMG stations in the study area, namely: Feldbach station (11298) and Bad Gleichenberg (11244), which measure precipitation with 1 min temporal resolution. These two stations were added to this step in September 2011.
2. Climatological scaling of the radar data using a climatological scaling factor to partially correct topographic shielding. The scaling factor is the ratio between the multi-year, 3-monthly accumulated precipitation from the station data and the corresponding accumulated radar precipitation data.
3. Rescaling radar data using the latest observations: based on the comparison of observations and radar fields at the station locations, the fields from the last step are rescaled again. The rescaling is a weighted average of the ratio between the data from the radar and the nearest rain gauge, where the weight decreases with increasing distance, increasing difference in climatological scaling, and decreasing rain at the station.





4. Final combination: The precipitation fields from steps 1 and 3 are combined into INCA fields through a weighting relationship, where the weight factor decreases with increasing climatological scaling. At the station locations, INCA is equal to the interpolated station field of step (1). Between the stations, the weight of radar information increases. In the areas where the radar return is weak due to orographic shielding, the analysis reduces to station interpolation, considering elevation effects.

Related to this study, it should be noted that the closest radars to the study area are Zirbitzkogel (approx. 100 km) and Rauchenwarth (approx. 140 km) (see Table B1 Appendix B). Considering these distances and the mountains between the study area and radars, the minimum detection height by the radar network in the study area is about 2000 m above the ground, leading to detection and estimation errors. Based on Kann et al. (2015), the ground clutter correction is the only correction of radar data. Hence, some errors such as bright band, signal attenuation, scan strategy, radar miscalibration, radome wetting, and errors due to non-meteorological echoes may still exist in INCA precipitation products.

Kann et al. (2015) used WegenerNet station data to evaluate 5 min INCA analysis data (rapid-INCA) for wet season (April-September) of 2011 and 4 different heavy precipitation events. The study showed a general underestimation in rapid-INCA during the wet season. The rapid-INCA also underestimated the average precipitation rate in three out of 4 events. They also showed the roles of rain gauges and radars in rapid-INCA analysis performance.

## 3 Methodology

### 3.1 Data Preparation

Precipitation data from 2007 to 2018 are used in this study. After transforming WegenerNet gridded data to the Lambert conformal conic projection, we used the conservative remapping scheme (Jones, 1999) to generate 1 km gridded fields (see Fig. B1 in Appendix B). The conservative remapping scheme is based on preserving water flux and has been widely used as a remapping scheme for precipitation observations and climate model outputs (e.g., O'Gorman, 2012, Nikulin et al., 2012, Sillmann et al., 2013, Prein and Gobiet, 2017, Tapiador et al., 2020, Fallah et al., 2020). We aggregated the 5 min WegenerNet data to 15 min to have the same temporal resolution in both datasets. As an example, Fig. 2 shows an intense precipitation event in May 2009 in INCA and WegenerNet gridded data with 15 min temporal resolution.


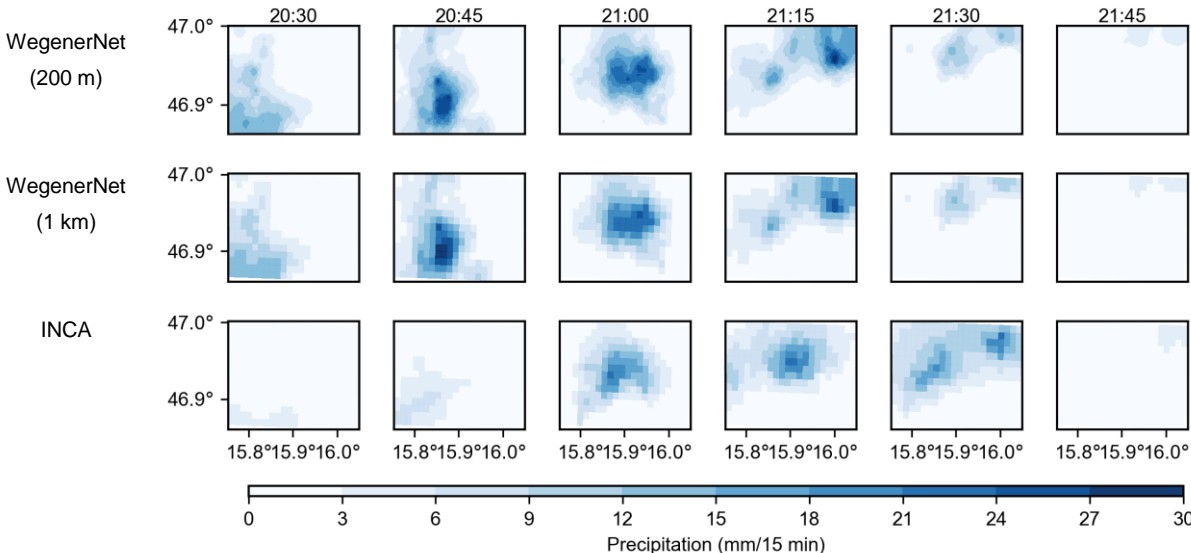

**Figure 2.** A convective event on 18 May 2009 that was observed by WegenerNet in original 200 m resolution (top) and 1-km resolution (center), and by INCA (bottom).

Although WegenerNet gridded data are generated from quality-controlled station data (Kirchengast et al., 2014), some errors may still remain in the data. The Mahalanobis Distance method – an approach to find multivariate outliers (Ben-Gal, 2005) – was implemented to detect these possible errors in WegenerNet gridded data. A brief explanation of this approach can be found in Appendix A. We only found a few grid cells that had to be treated as outliers for parts of the measurement period (see Table A1 Appendix A): Station 44 is peculiar because it is located on top of a tall building, at 55 m above ground, and therefore suffers from stronger wind-induced undercatch. Station 145 is another outlier, which has been detected by the WegenerNet team in 2015, which led to a replacement of its rain gauge. Since we found only few outliers, it can be concluded that the operational quality control system of the WegenerNet can filter outliers reasonably well.

## 3.2 Approaches

We use the annual area-mean precipitation and compare it in both datasets to gain a general overview of INCA developments and find possible systematic errors. The annual difference between the two datasets in each pixel is also calculated to portray the spatial pattern and the possible relation with ZAMG stations' distance. This is supported by a comparison of the INCA, ZAMG station, and WegenerNet data at the Feldbach station location (hereafter Feldbach cell). For WegenerNet values, we used the average of the four closest stations to the Feldbach cell (station numbers: 45, 46, 60, and 61), since there is no WegenerNet station located in this cell. As mentioned in Sect. 2.1, most of the precipitation occurs from May until September (hereafter wet season) due to a high proportion of convective events, and the other months (October to April) have noticeably less precipitation (hereafter dry season). To determine possible dependencies on the seasonality, we separately compare INCA and WegenerNet in the wet and dry seasons, respectively.



Since the Austrian weather radars overlook events that happen lower than about 2000 m above the ground in this region (due to radar beam blockage by the surrounding mountains) and the radar data are only corrected for ground clutter, some detection errors such as missing events and classifying non-precipitation phenomena as precipitation can occur. We implement three indices: Probability of Detection (POD), False Alarm Ratio (FAR), and Critical Success Index (CSI) to address these errors in INCA. We calculate POD and FAR indices for each grid cell, classifying them into seven different

precipitation intensities from 0.1 to 5 mm per 15 min. We then illustrate them for each pixel in order to analyze possible dependencies on the distance from the two ZAMG stations.

One way to address extreme precipitation is to consider the higher part of the intensity distribution (e.g., 99th quantile) as a threshold and calculate the average intensity of all time steps with higher intensities (the highest 1 %). The benefit of using this approach is that it includes changes in all events above the threshold (Haylock and Nicholls, 2000). We consider all

events with equal or more than 0.1 mm per 15 min that happened in both datasets simultaneously and then calculate the average intensity of the highest 1 % for each dataset.

Since event-based analysis is important for some hydrological studies, such as soil erosion and runoff generation, we separate events in each dataset and evaluate INCA's ability from this perspective. We use a simple threshold based on the minimum dry period between two consecutive events (Minimum Inter-event Time MIT) to separate individual events. This

approach has been used in different studies (e.g., Brown et al., (1985); Haile et al., (2011)) to identify individual events. A wide range of MIT values has been selected in the literature. Choosing different MIT values leads to different characteristics of derived events. There should be a compromise between the independency of events and intra-event variability (Dunkerley, 2008). Since the study is affected by both convective and large-scale systems, we choose the MIT value to be 1 hour and the minimum precipitation to be 0.1 mm per 15 min at any pixel. After separating events based on these criteria for each dataset,

we analyze the characteristics of these events, such as event duration, accumulated precipitation, area-average intensity, peak intensity ,and the average number of wet cells, both in INCA and in WegenerNet. Note that the accumulated precipitation is the total amount of area-average precipitation during an event (mm), area average intensity is the total precipitation divided by the duration of an event (mm h$^{-1}$), peak intensity is the maximum intensity at a pixel during an event (mm per 15 min), and the average number of wet cells is the average number of cells that have more than 0.1 mm per 15 min of precipitation

during an event.

To study extreme convective short-duration events (ECSDEs) based on the events in the previous paragraph, we define an ECSDE by having three characteristics: 1. the area average intensity is more than the 95[th] quantile of WegenerNet intensities, 2. the duration is less than four hours, and 3. a coverage of less than 2/3 of the study area. We evaluate INCA's performance in these events with a focus on the spatial characteristics of 4 ECSDEs in both datasets, along with the area-mean and peak

intensities.



## 3.3 Comparison Metrics

Error metrics have been widely used in many different disciplines, such as hydrology and hydrometeorology, to quantify model result accuracies or compare observations and forecasts. Each metric has its own strengths and weaknesses and quantifies a different aspect of the model. Hence, using multiple metrics for the comparison is more desirable (Jackson et al., 2019). We use four common metrics: Bias, Relative Difference (RD), Root Mean Square Error (RMSE), and Correlation Coefficient (CC), for comparing INCA precipitation estimates with WegenerNet. These indices are computed according to Eq. (1) to (4) in Table 1 below.

**Table 1.** Comparison indices used in this study and their equations (partly adopted from Jackson et al., (2019))

| Number | Name | Equation | Range |
|---|---|---|---|
| (1) | Bias | $\text{Bias} = \frac{1}{N} \sum_{i=1}^{N} RR_i - WR_i$ | $-\infty$ to $+\infty$ |
| (2) | Relative Difference | $RD = \frac{\sum_{i=1}^{N}(RR_i - WR_i)}{\sum_{i=1}^{N} WR_i} \times 100$ | $-\infty$ to $+\infty$ |
| (3) | Root Mean Square Error | $RMSE = \sqrt{\sum_{i=1}^{N} \frac{(RR_i - WR_i)^2}{N}}$ | $0$ to $+\infty$ |
| (4) | Correlation Coefficient | $CC = \frac{\sum_{i=1}^{N}(RR_i - \overline{RR})(WR_i - \overline{WR})}{\sqrt{\sum_{i=0}^{N}(RR_i - \overline{RR})^2}\sqrt{\sum_{i=0}^{N}(WR_i - \overline{WR})^2}}$ | $-1$ to $+1$ |

Where $RR$ and $WR$ indicate the precipitation estimates by INCA and by WegenerNet, respectively; $i$ is the index of the time step, $N$ is the number of time steps and the top bar shows the average over time.

Additionally, in order to evaluate INCA's ability to detect precipitation, we analyzed three categorical indices: Probability of Detection (POD), False Alarm Ratio (FAR), and Critical Success Index (CSI), based on the equations in Table 2 below.



**Table 2.** Detection indices used in this study and their equations

| Number | Name | Equation | Range | Note |
|:------:|:----:|:--------:|:-----:|:-----|
| (5) | POD | $\text{POD} = \dfrac{a}{a+c}$ | 0 to 1 | 1 indicates the best performance of precipitation detection |
| (6) | FAR | $\text{FAR} = \dfrac{b}{a+b}$ | 0 to 1 | 0 indicates the best performance of precipitation detection |
| (7) | CSI | $\text{CSI} = \dfrac{a}{a+b+c}$ | 0 to 1 | 1 indicates the best performance of precipitation detection, a combination of POD and FAR |

Where $a$ is the number of events that both datasets could capture, $b$ is the number of false events detected by INCA, and $c$ is the number of missed events.





# 4 Results and discussion

## 4.1 Annual precipitation

The area-mean of annual precipitation in INCA and WegenerNet datasets, and the average of WegenerNet stations are shown in Fig. 3a. The annual average of precipitation in the 12 years from 2007 to 2018 is 979 mm in INCA and 881 mm in WegenerNet. Overall, INCA area-mean values exceed WegenerNet values with an average relative difference of 11 %. The only year that INCA underestimates precipitation is 2009, a year with particularly high precipitation: At the long-term ZAMG station Bad Gleichenberg, it was the wettest year since 1937. From 2012 to 2014, the overestimation by INCA is even larger, with an average relative difference of 24.5 %. It is worth noting here that the difference between the average precipitation of WegenerNet stations and the area-mean gridded data is negligible (0.15 % on average).

Annual precipitation in the Feldbach cell is shown in Fig. 3b. Similar to the annual area-mean value, INCA also overestimates precipitation in this cell, but to a smaller amount, with an average relative difference of 3 %. In this cell, from 2007 until 2014, INCA overestimated precipitation with an average of about 5 % and a maximum of 13 % in 2012. Starting from 2015, INCA tends to underestimate precipitation by an average of 1.3 % and a maximum of 2.8 % in 2016. Compared to the annual-mean values, INCA performs better in this cell, indicating the importance of the rescaling step (step 3 in Sect. 2.2) in the INCA estimates. Also, INCA values are consistently higher than those of the ZAMG station Feldbach with an average of 7 %. As shown in Fig. 3b, INCA has considerably higher values than Feldbach station in 2012 and 2013 with a relative difference of 21 % and 17 %, respectively. It is worth mentioning that the Feldbach station has lower values than WegenerNet except for 2007, 2009, and 2010. Based on these results, we can conclude that using the merged radar-gauge data leads to more accurate estimates than individual ZAMG station data, even at the ZAMG station location. Furthermore, and similar to Fig. 3b, we show the annual precipitation in the cell where station Bad Gleichenberg is located (Fig. B2 Appendix B). Comparing to station Feldbach, the difference between INCA and WegenerNet is higher, which could be due to the weight of this station in step 1 (see Sect. 2.2).

Figure 4 shows annual maps of the relative differences between INCA and WegenerNet. Similar to Fig. 3, there is a considerable overestimation in INCA from 2012 to 2014, especially in the cells that are far from ZAMG stations. Since there was a replacement period of the Austrian radars (starting in October 2011, details in Table B1, Appendix B), and because the closest radar (Zirbitzkogel) was off during two periods of time (June 2012-October 2012 and February 2013-March 2013), we interpret this overestimation as an error of the radar data. This error was partly removed close to the ZAMG stations by step 4 of the INCA data preparation (see Sect. 2.2).





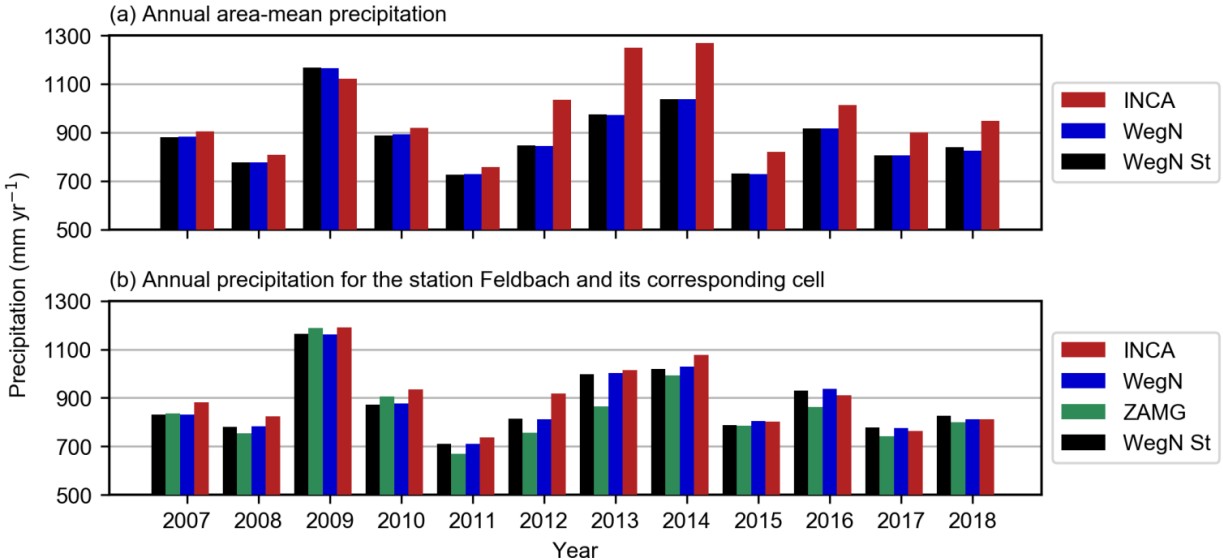

**Figure 3. (a)** Annual area-mean precipitation. **(b)** Annual precipitation for the ZAMG station 11298 (Feldbach) and its corresponding cell.

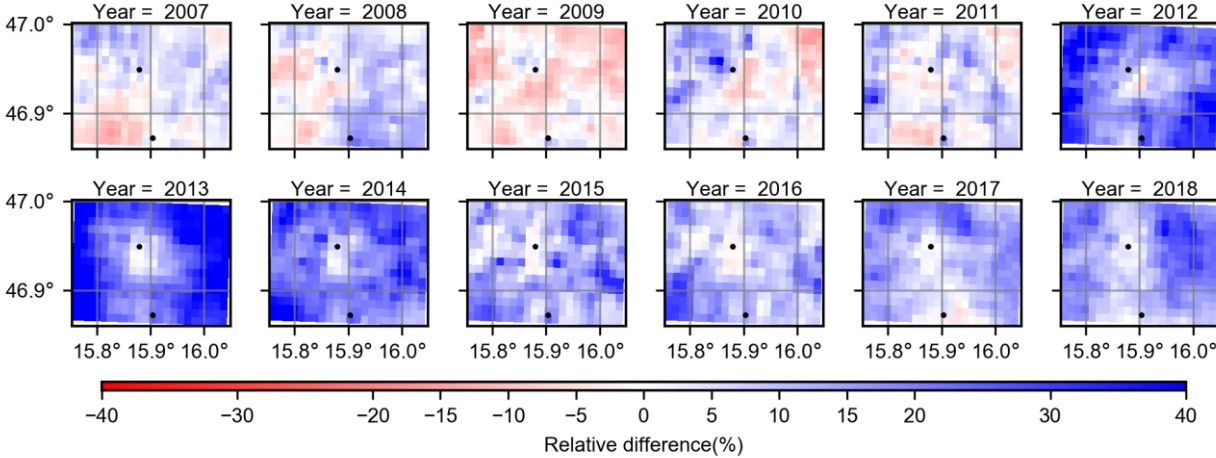

**Figure 4.** Annual relative difference between INCA and WegenerNet in each pixel. The black dots indicate the two ZAMG stations in the study area.

As shown in Fig. 4, the INCA performance can be divided into three different periods: 2007-2011, 2012-2014, and 2015-2018. From 2007 to 2011, INCA generally overestimated precipitation, except for 2009. In general, the annual area-mean differences differ between -3.9 % and +4.3 %, with an average of 1.9 % in this period. In addition, there was no specific pattern close to the two ZAMG stations since these stations were added to the INCA algorithm in September 2011. Although the relative differences of area-mean values are positive in this period except 2009, INCA underestimates precipitation in some cells.





From 2012 to 2014, INCA considerably overestimated precipitation in almost all grid cells, and the annual area-mean
difference rose to almost 29 % in 2013. As shown, relative differences are lower at the cells closer to ZAMG stations,
especially around Feldbach station. We interpret this as an error, introduced by the new radar, which was partly removed by
the calibration with ZAMG station data.

From 2015 to the end of 2018, the overestimation by INCA was still predominant. The overestimation, however, is smaller
compared to the previous period. Nevertheless, annual-mean differences are almost always positive, with typical values
between +11 % and +15 % and an average of +12.5 %. This overestimation is smaller in the grid cells near the two ZAMG
stations, indicating again the role of these stations to adjust radar data in the INCA algorithm.

Note that we also calculate RMSE for the annual precipitation (see Fig. B3 Appendix B). There is no noticeable spatial
pattern in the first period, and the minimum and maximum of area-mean annual RMSE error are 45.0 mm and 57.5 mm,
respectively. This error is considerably higher in the second period with a minimum and maximum of 191.0 mm and
278.0 mm, respectively. In this period, RMSE is lower close to the Feldbach station (just like the relative difference in Fig.
4). Similar to the relative difference, the area-mean annual RMSE decreases in the 2015-2018 period. For this period, the
minimum RMSE is 92.6 mm in 2015 and the maximum RMSE is 124.1 mm in 2018.

As mentioned before, more stations were added to the INCA algorithm during the years. To study possible changes on INCA
estimates after adding these stations, the raw radar data used in INCA should be included in future analyses.

**4.2 Precipitation detection**

Figure 5 shows POD and FAR values for different thresholds for each cell for the three different time periods identified
above. The dashed black and solid black lines indicate POD and FAR for the Feldbach cell. As can be seen, INCA's ability
to detect precipitation decreases at higher intensities. The change of POD and FAR in the Feldbach cell is different. At
thresholds lower than 0.5 mm per 15 min, it has lower POD than other cells, especially since 2012. The FAR value performs
better compared to other cells at lower thresholds. This could be due to the INCA algorithm for removing false precipitation
events, which unintentionally removes some light precipitation events.

Generally, INCA has the highest POD values at a threshold of 0.1 mm per 15 min. With increasing precipitation intensity,
POD tends to decrease. The FAR values also have the best performance at 0.1 mm per 15 min and start to increase with
higher intensities. When longer time intervals (e.g., one hour, three hours, and six hours) are considered, the detection errors
decrease (not shown).



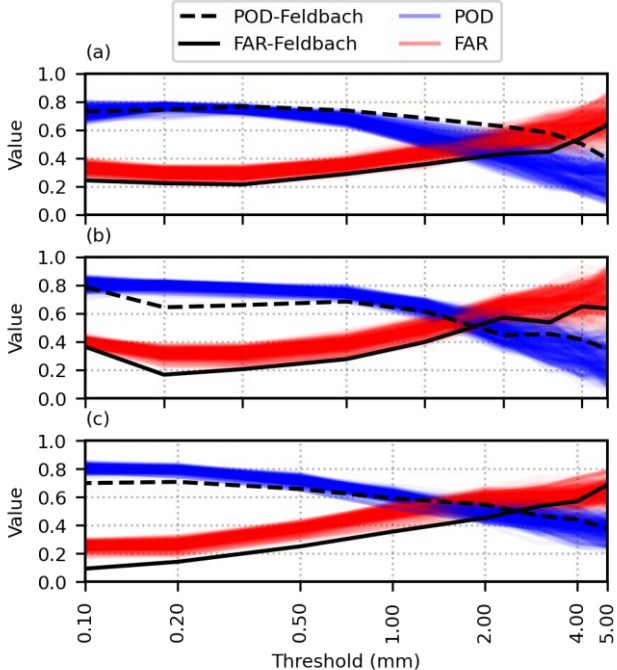

**Figure 5.** POD and FAR values based on different thresholds for each cell for 2007 – 2011 (top), 2012 – 2014 (center), and 2015 – 2018 (bottom). Note that the dashed black and the black lines indicate POD and FAR for the Feldbach cell, respectively.

Figure 6 shows POD and FAR for each grid cell based on the 0.1 mm per 15 min threshold for the three periods, separated in wet and dry seasons. The FAR performs better in the grid cells near the two ZAMG stations. POD, however, has lower values in those cells. As previously mentioned, this may be due to the INCA algorithm, which unintentionally removes some light events. The ability of INCA in detecting precipitation is noticeably higher in wet seasons.

We also calculated CSI for each cell (Fig. B4 Appendix B). Since CSI is a combination of POD and FAR, its behavior is a combination of POD and FAR; i.e., CSI performs better during the wet season. CSI values are higher in the third period (2015-2018) for the dry seasons. Since the weight of the radar estimate becomes higher with increasing distance from the ZAMG stations in the INCA algorithm (see Sect. 2.2), we conclude that the radar detected some precipitation events, which were not observed by ground stations. Since the radar sees precipitation in the study area only beyond about 2000 m above the ground and some errors are not corrected (see Sect. 2.2), 'false events' can be due to events that do not reach the ground due to evaporation or due to non-precipitating phenomena. The latter error can explain higher FAR values in the cells with longer distances from ZAMG stations.





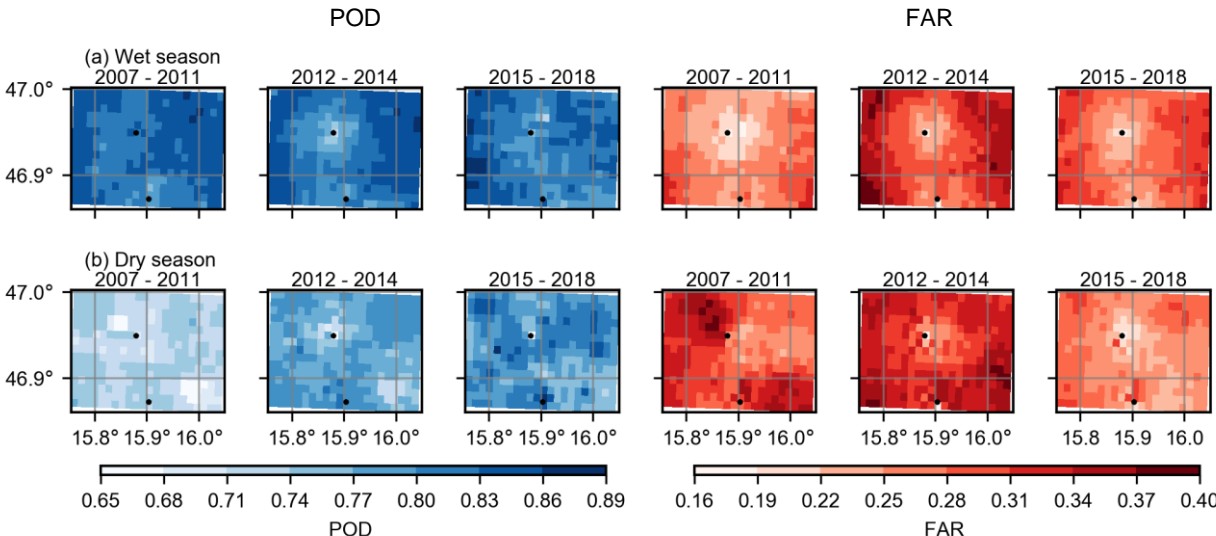

**Figure 6.** POD and FAR for 15 min data based on the 0.1 mm threshold for three different periods of time for wet (top) and dry (bottom) seasons. The black dots indicate the two ZAMG stations.

## 4.3 Seasonal comparison

Figure 7 displays the mean relative differences between INCA and WegenerNet in the wet (May - September) and dry seasons (October – April) for the three periods. In contrast to the relative differences in Fig. 4, these differences were calculated for events with precipitation of equal or more than 0.1 mm per 15 min that happened in both datasets. The 0.1 mm threshold is implemented due to the resolution of the tipping bucket gauges used in WegenerNet. In the first period, both overestimation and underestimation can be detected in INCA grid cells. Starting in 2012, overestimation is dominant in most INCA grid cells, similar to annual precipitation. This overestimation is higher in the wet seasons and grid cells farther away from the two ZAMG stations. We can conclude that the new radar settings (cf. Sect. 2.2) tend to overestimate precipitation and that the INCA algorithm works reasonably well closer to the two ZAMG stations. However, the overestimation farther away from those stations is still considerably large. The higher relative difference in wet seasons is an indication of difficulties in the radar network to estimate intense rainfall events. Considering RMSE, the pattern is similar to the relative difference (not shown).

Furthermore, we calculate the temporal correlation coefficient between INCA and WegenerNet (Fig. B5 Appendix B). Based on these results, the correlation is noticeably lower in the wet seasons. We interpret this as a consequence of a higher percentage of convective events, which are harder to capture. Similar to the relative difference, INCA performs better in the cells close to the ZAMG stations in the wet seasons.





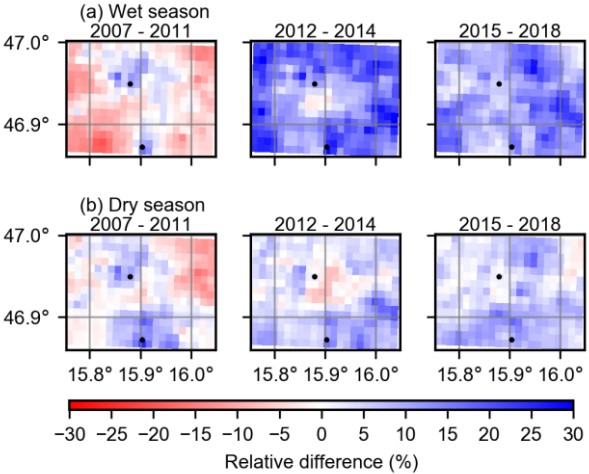

**Figure 7.** Mean relative difference between INCA and WegenerNet for 2007-2011 (left), 2012-214 (center) and 2015-2018 (right) in dry (top) and wet (bottom) seasons. Note that only events with equal or more than 0.1 mm per 15 min were considered.

## 4.4 Extreme precipitation

In this section, we compare extreme events in INCA and WegenerNet based on the different seasons for the three periods. Note that the 99th quantile was calculated for time steps with precipitation equal to or more than 0.1 mm per 15 min, which happened in both datasets (see Sect. 3.2). Figure 8 shows the mean values of all time steps exceeding quantile 99 in each pixel for both datasets and the relative differences in wet and dry seasons.

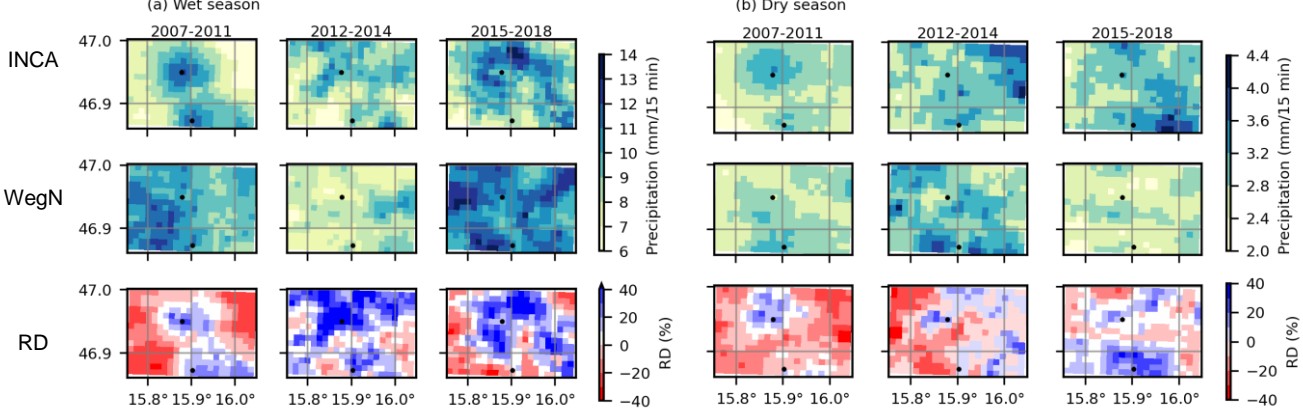

**Figure 8.** Spatial patterns and relative difference of the mean of all time steps exceeding quantile 99 for INCA and WegenerNet for **a)** wet and **b)** dry seasons in the first (left), second (middle), and third (right) period



From 2007 to 2011, INCA tends to overestimate extreme precipitation in cells close to the two ZAMG stations and underestimated it in the other parts of the study area, similar to the mean seasonal values (see Fig. 7). Compared to the mean seasonal values, underestimation is larger in extreme precipitation, especially in the dry season. The maximum and minimum differences in this period are 26 % and -45 %, respectively. It is worth mentioning that the spatio-temporal evolution of extremes is not particularly well captured by INCA (cf. Fig. 2).

Between 2012 and 2014, an overestimation of INCA in the corresponding cells of ZAMG stations is also noticeable, particularly in the wet season. In contrast to annual and mean-seasonal values, there is no relationship between distance from ZAMG stations and the relative difference. In the dry season, INCA shows less underestimation than in the first period. The maximum and minimum differences in this period are 101 % and -36 %, respectively. For wet seasons in the 2015-2018 period, the behavior was relatively similar to the second period with a decrease in overestimation. In this period, the

maximum and minimum differences are 44 % and -29 %, respectively. Based on these results, the overestimation of INCA is larger in extremes, especially in the wet seasons.

### 4.5 Event-based evaluation

In this section, we consider individual events, and based on the criteria we described in Sect. 3.2, we identified 4699 separate events in INCA and 5116 in WegenerNet over 12 years. The number of events in different seasons is shown in Fig. 9.

Similar to Sect. 4.2, the number of detected events in INCA is lower in the dry seasons. Note that the number of dry-season-events can be slightly biased in WegenerNet, due to snowfall events, which can get recorded twice: Once when the snow is measured by heated rain gauges, and again when the snow melts at the unheated gauges. In general, snowfall events in the region are rare (cf. Sect. 2.1); thus we do not expect them to have significant influence in metrics other than the number of events. Another effect on the number of dry season events is that the radars tend to miss precipitation more often in winter

due to beam blockage by surrounding mountains (cf. Sect. 2.2), especially for low lying clouds which are often present in the dry season.

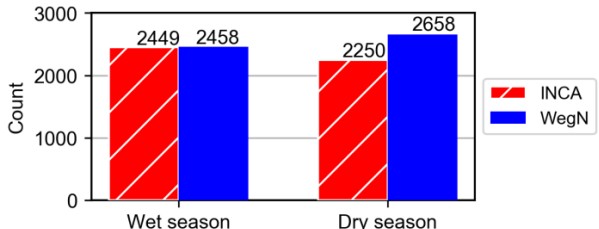

**Figure 9.** Number of events in different seasons in both datasets

Table 3 describes the statistics of separate events in both INCA and WegenerNet. Note that the accumulated precipitation and precipitation rate are based on the area-mean value in each time step. The peak intensity, however, is the maximum value that happened in one cell during an event.





**Table 3.** Descriptive statistics for separate events in INCA and WegenerNet based on the 1-hour interval (based on definitions in Sect. 3.2).

| | Duration (min) | | Accumulated precipitation (mm) | | Area-average intensity (mm h$^{-1}$) | | Average number of wet cells | | Peak intensity (mm 15 min$^{-1}$) | |
|---|---|---|---|---|---|---|---|---|---|---|
| | INCA | WegN | INCA | WegN | INCA | WegN | INCA | WegN | INCA | WegN |
| mean | 172 | 175 | 2.28 | 1.90 | 0.37 | 0.25 | 63 | 41 | 1.67 | 1.54 |
| 25% | 15 | 15 | 0.006 | 0.001 | 0.01 | 0.002 | 8 | 2 | 0.16 | 0.13 |
| 50% (median) | 75 | 60 | 0.07 | 0.01 | 0.06 | 0.01 | 28 | 8 | 0.34 | 0.23 |
| 99% | 1305 | 1665 | 32.13 | 29.18 | 3.76 | 2.7 | 288 | 241 | 20.99 | 18.30 |
| max | 2460 | 4065 | 101.63 | 105.86 | 13.31 | 9.30 | 330 | 313 | 46.17 | 35.04 |


Based on Table 3, the average accumulated precipitation and the precipitation rate measured by INCA are higher than WegenerNet. Similarly, the average number of wet cells is higher in INCA, which can affect the accumulated precipitation. The difference between the average number of wet cells in INCA and WegenerNet is higher in the dry season. This could be due to a slightly lower effective resolution of INCA in the study area, where the radar beam of the nearest radar is already

comparatively wide. Whereas the average duration of events only differs by 3 min, the difference increases significantly for longer events.

To check a possible time shift in INCA, we consider events that fulfill the following conditions: 1. The absolute difference between the starting time of an event in INCA and WegenerNet is less than 1 hour, and 2. the absolute difference between the ending time of an event in INCA and WegenerNet is less than 1 hour. Based on these criteria, INCA started earlier to

detect precipitation in 40 % of these events with an average of 35 min. Both datasets started at the same time for 24 % percent of events, and for the rest (36 %), INCA started later by an average of 41 min. For the ending of an event, 55 % of events end later in INCA by an average of 37 min and 25 % end earlier with an average of 44 min. Note that, starting / ending time is considered within 15 min since the temporal resolution is 15 min.

To separate the errors associated with time shift from errors related to intensity, we focus only on those events that happened

at the same time in both datasets and we found 2949 events. Similar to the results in Table 3, the accumulated precipitation is higher in INCA and the bias value is 0.14 mm 15 min$^{-1}$. Although INCA overestimates accumulated precipitation most of the time, the peak intensity is slightly higher in WegenerNet except during July. The overall bias for peak intensity is -0.04 mm 15 min$^{-1}$.

We also studied the time of the peak intensity in both datasets and found that the peak happens during the first half of the

event duration. Also, in the majority of events, the peak intensity in INCA happens slightly later (approximately 5 min) than in WegenerNet.

The monthly average of the peak intensity is shown in Fig. 10. INCA generally underestimates the precipitation peaks in the first period but generally overestimates them from mid-2012 onwards. This behavior is likely due to the change of the radar





network in 2012. There is a noticeable peak-intensity overestimation in mid-2013. Contrary to the mean precipitation (see Fig. 3a and 4), the differences in peak-intensity between INCA and WegenerNet decreased significantly in 2018.

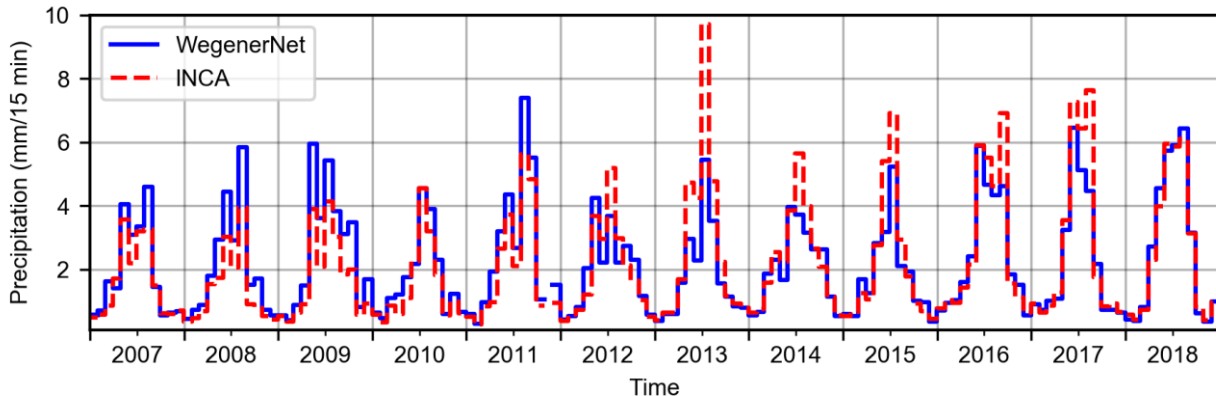

**Figure 10.** Monthly mean of the peak (mm per 15 min) for each event for WegenerNet and INCA

### 4.5.1 Extreme convective short-duration events

Based on these events and the criteria described in Sect. 3, there are 62 extreme short-duration convective events (ECSDE) detected by WegenerNet during the 12 years study period. Among these 62 events, 47 are detected by INCA. The same underestimation pattern before 2012 and overestimation afterward (see Fig. 10) can be seen in accumulated precipitation and peak intensity. The highest underestimation of accumulated precipitation happened in 2009 (up to 40 %). For the peak intensity, INCA underestimates it by up to 60 % in 2008 and overestimates by approximately 65 % in 2015. The results are similar to the extremes in Sect. 4.4.

To check the ECSDE's spatial patterns in both datasets, we focus on four examples of these events. As shown in Fig. 3a and discussed in Sect. 4.1, the precipitation amount was considerably higher in 2009. We choose the first ECSDE event from this year, which has the highest peak intensity in 2009. The second event in 2011 is special because INCA overestimated accumulated precipitation but underestimated the peak intensity. The third and fourth events are selected from the third period (2015-2018) with different characteristics to show INCA's performance based on the latest improvements. The third event has the highest maximum intensity in INCA, and the fourth event's peak intensity is highest in this period (2015-2018).





Event on 18 May 2009: In Fig. 2, we already showed the difference in the spatial structure of this extreme event on May

18th, 2009, in both INCA and WegenerNet. Based on WegenerNet data, the accumulated rainfall over the entire study area

was 23.7 mm in 2 hours, but the accumulated rainfall that happened in the wettest cell was 46.6 mm. As shown in Fig. 11,

the WegenerNet detected two peaks for this event at 20:45 and 21:15. In INCA, however, the two peaks happened at 21:00

and 22:00. It is worth mentioning that the location of the peak(s) is also different in INCA (see Fig. 2). The area-mean of

rainfall shows a time-shift in INCA of 30 min. In addition, rainfall starts and ends earlier in INCA, and INCA has more wet

cells during this event.

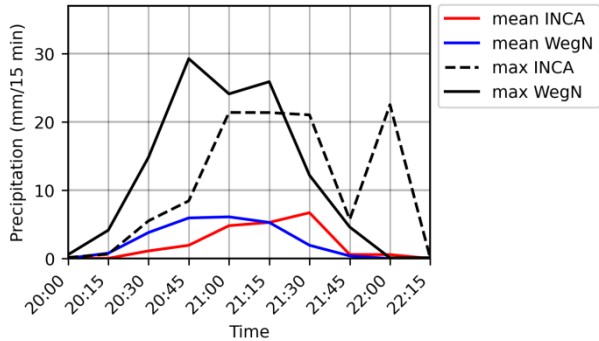

**Figure 11.** Time series of area-mean and maximum values in INCA and WegenerNet for the event on 18 May 2009.

Event on 1 September 2011 (Fig. 12): During this event, an areal average of 27.6 mm fell in 3.25 hours (50 mm in the

wettest cell). There is a time shift in the peak intensity and area average rainfall. Contrary to the event in 2009, the peak

intensity happened one time step earlier in INCA. The differences between INCA and WegenerNet values are smaller than in

the previous event. INCA starts to detect earlier and finishes later than WegenerNet, and INCA has more wet cells.

Comparing these results with those by (Kann et al., 2015), we see that the 15 min INCA precipitation analysis performs

better than 5 min rapid INCA. The time shift in detection was also observed in rapid-INCA, caused by radar data.






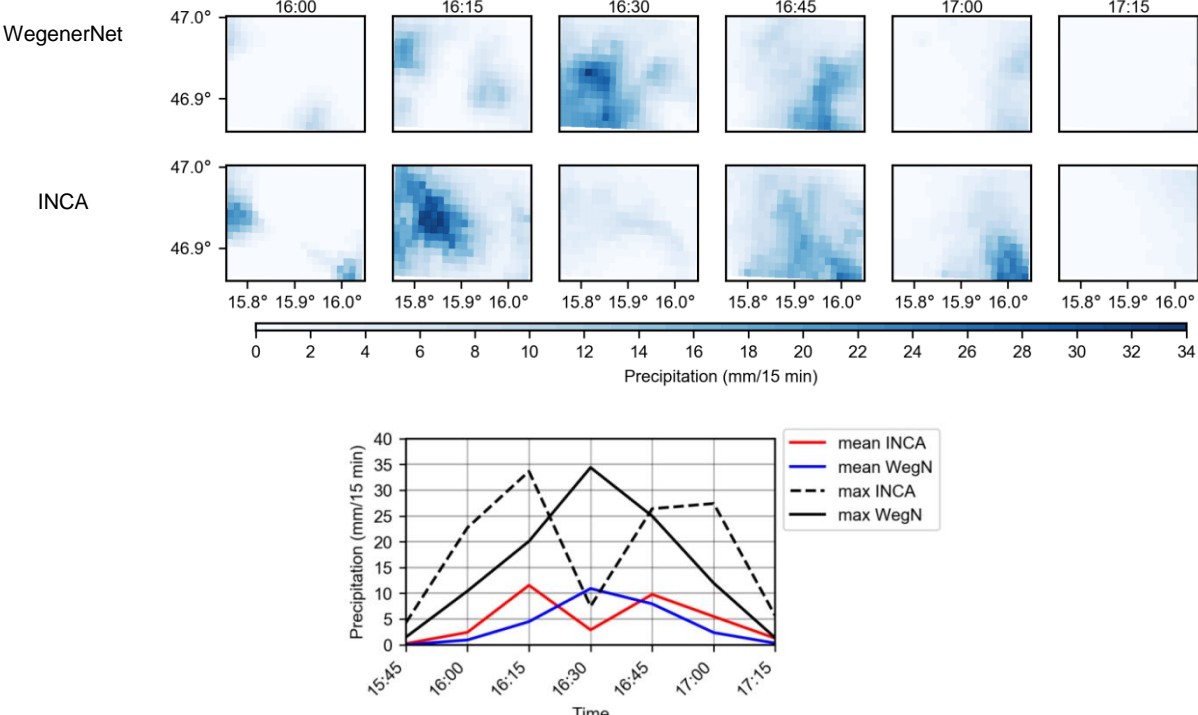

**Figure 12.** An event on 1 September 2011 in WegenerNet (top) and INCA (center), and time series of area-mean and maximum values in INCA and WegenerNet (bottom)

Event on 16 August 2016 (Fig. 13): Based on WegenerNet data, the areal average of the total amount of rainfall was 19.4 mm, which fell in 2.25 hours (36.5 mm in the wettest cell). INCA overestimated the event peak by about 40 % in this event. Similar to the event in 2009, there is a time lag in INCA's detection of the rainfall peak (Fig. 13). The highest value of the area-mean rainfall happened at 21:30 in both datasets. This event starts earlier and finishes later in INCA, with more wet cells in WegenerNet.


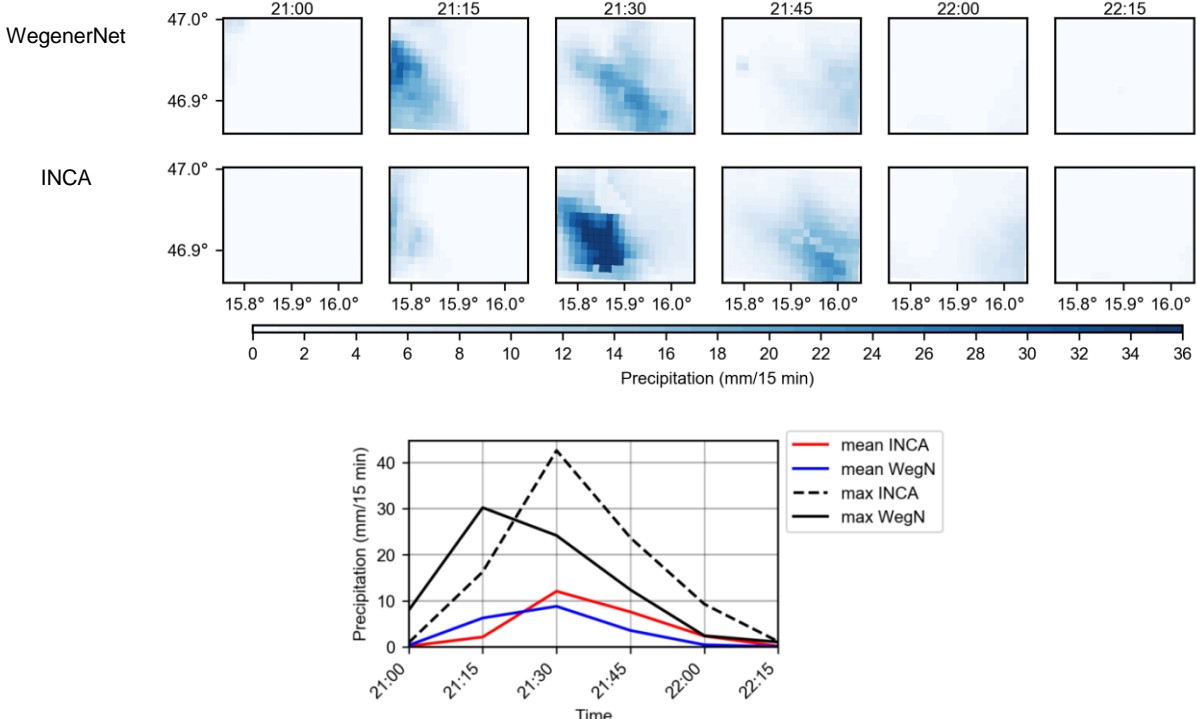

**Figure 13.** An event on 16 August 2016 in WegenerNet (top) and INCA (center), and time series of maximum and area-mean values in INCA and WegenerNet (bottom)


Event on 15 July 2018 (Fig. 14): The areal average of accumulated rain was just 10.8 mm that fell in 2.5 hours. However, the accumulated rain in the wettest cell was 59.6 mm in this event. INCA detected the peak intensity one time step later and had lower values than WegenerNet. The maximum value of rainfall in this event happened at 14:00 in WegenerNet, and INCA significantly underestimates this value (Fig. 14). It is worth mentioning that the location of the maximum value is far from
the two ZAMG stations. This event starts and ends earlier in INCA with a higher number of wet cells.

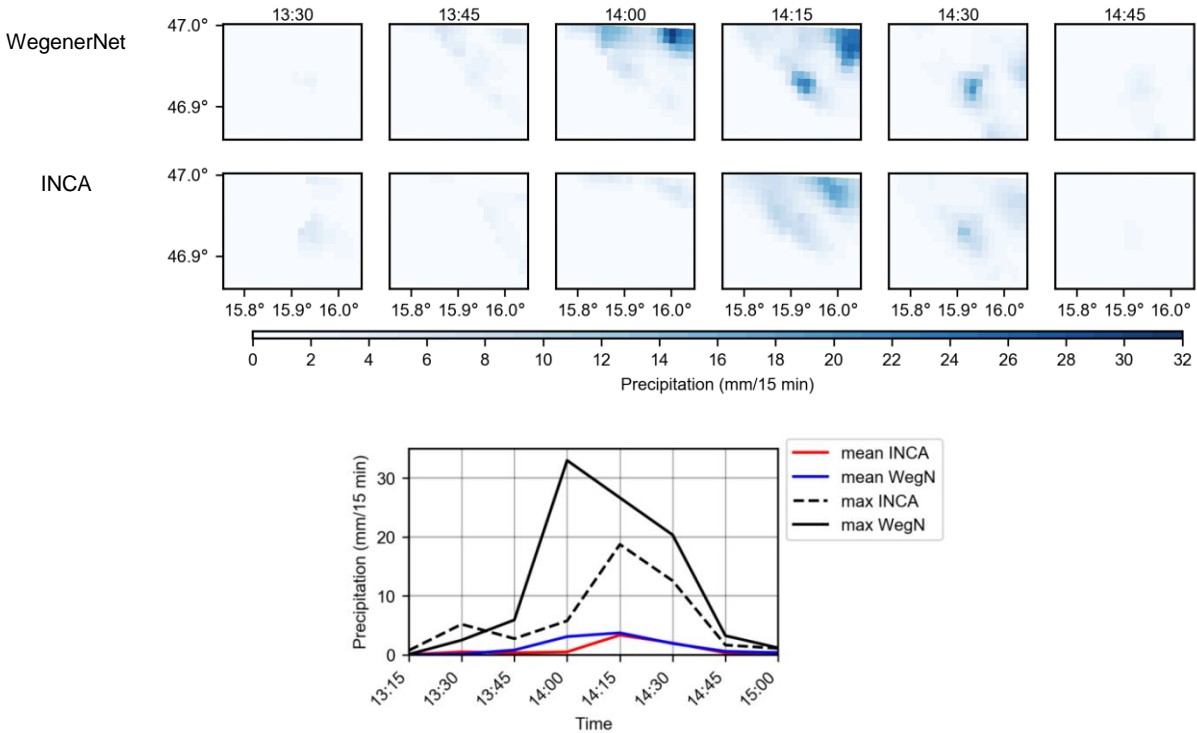

**Figure 14.** An event on 15 July 2018 in WegenerNet (top) and INCA (center), and time series of maximum and areal mean values in INCA and WegenerNet (bottom).


INCA detects the peak intensity later than WegenerNet, except for the event on September 1st, 2011. INCA underestimates the peak intensity in two events (2009 and 2018), and overestimates this value in the 2016 event. The overestimation and underestimation is more pronounced for peak intensities than for area-mean values. These error and time-shifts in detection may affect the performance of flood warning systems.

**5 Conclusions**

The evaluation of precipitation estimates helps to improve the understanding of errors and uncertainties from different sources (e.g., systematic errors, random errors, and spatio-temporal dependency). In this study, we evaluated INCA (Integrated Nowcasting through Comprehensive Analysis) precipitation analysis products of the ZAMG (the Austrian Central Institute for Meteorology and Geodynamics), using WegenerNet high-resolution gridded data from 2007 to 2018 in

southeast Austria. First, we compared annual precipitation estimates of INCA and WegenerNet for each pixel and the area-mean values. In general, INCA overestimates the area-mean annual precipitation except in 2009, which was a particularly wet year. The performance of INCA can be divided into three different periods; from 2007 to 2011, both overestimation and underestimation are observed in INCA, with more pronounced underestimation in the wet season. Starting in 2012, INCA





considerably overestimates precipitation by up to 60 %. However, this overestimation is less pronounced close to the station

Feldbach. Starting in 2015, this spatial pattern continues but with a lower overestimation compared to the second period. We conclude that this overestimation is a result of systematic errors from newly installed radars. This overestimation was partly removed in the INCA algorithm using reference gauges.

We used categorical metrics to study the ability of INCA in detecting precipitation. Generally, the number of false events is smaller in the cells closer to ground stations operated by the ZAMG (which are used as input for INCA), especially in the

wet season. Surprisingly, the number of true events close to the ZAMG stations is comparably smaller too. This could be due to the INCA algorithm for removing false precipitation events that unintentionally removes some light precipitation events. We evaluated extremes during these three periods, and for wet and dry seasons. In the first period, INCA overestimates precipitation in cells close to the ZAMG stations and underestimates it in other cells, especially during the wet season. This overestimation is more noticeable, and dominates most of the study area in the wet season from 2012 until 2014. However,

for the dry seasons in this period, underestimation by INCA is dominant. For the third period, the pattern for the wet seasons is similar, but INCA tends to overestimate extremes during the dry seasons.

We also considered individual events in both datasets and analyzed their characteristics. Based on these results, INCA tends to underestimate the peak precipitation intensity of the events until mid-2012 and overestimates it afterward. The largest overestimation of the peak-intensity happened in July. Generally, the precipitation rate is higher in INCA, and there is a time

shift for event detection in INCA. Based on our results, INCA starts to detect precipitation earlier than WegenerNet in 40 % of the events and ends detecting it later than WegenerNet in 50 %. Considering four examples of extreme short-duration convective events, there is a time-shift in detecting the peak intensity in INCA. In these events, the peak intensity bias is considerably larger than in all events. In general, INCA has been improving in detecting and estimating precipitation. However, there are errors due to radar estimates and the algorithm for merging radar and rain gauges, which can negatively

affect the INCA analysis product. In addition, it is shown that gauges are crucial for correcting some errors due to radar estimates. Careful consideration must be taken when using merged rain-gauge–radar products, especially in extreme events.

**Suggestions for future studies:**

For future studies, it is suggested to include the raw radar data used in INCA in the analysis, to separate errors due to radar estimates. Also, we suggest using the results of this study to consider high-impact events and analyze the effects of INCA

uncertainties on risk management. In addition, the relation between wind speed and precipitation estimates in both gauges and radars needs to be considered separately.





## Appendix A

The Mahalanobis distance in matrix notation is calculated by the formula below:

$$D^2 = [x - m]^T C^{-1} [x - m]$$

A. (1)

Where $x$ is a data vector, $m$ is the multivariate mean, and $C$ is the estimated data covariance matrix (Filzmoser, 2016). After calculating Mahalanobis distance for each year, we compute the 99.9%-Quantile of the Chi-Square distribution with 2 degrees of freedom. Based on this, cells that have Mahalanobis distances above 13.8 are considered as outliers. The numbers and location of these outliers are shown in Table A1.

The cell with the coordinates ($x = 7$, $y = 10$) underestimated rainfall and was considered as an outlier in 2010. This cell is the corresponding cell of station 44, which is located on the top of a tall building. Furthermore, the cell with the coordinates ($x = 11$, $y = 0$) underestimated rainfall and was considered as an outlier in 2011. This cell refers to station 145. Starting from 2015, the Mahalanobis values for this cell were below the threshold, which is the year when the station has been replaced.

**Table A1** The numbers and locations of WegenerNet outliers based on Mahalanobis approach

| Year | Number of outliers | Locations (x, y) |
|------|-------------------|------------------|
| **2007** | 1 | (15,0) |
| **2010** | 2 | (7,10) (6,10) |
| **2011** | 2 | (11,0) (1,5) |
| **2015** | 1 | (1,14) |
| **2017** | 1 | (1,5) |



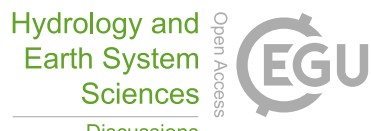

**Appendix B**

**Table B1** Location and the distance of the Austrian radars from the Feldbach station (station 11298)

| Name | Longitude | Latitude | Height of antenna (m MLS) | Radar replacement date | Distance from Feldbach station (km) |
|---|---|---|---|---|---|
| Rauchenwarth | 48.076 | 16.535 | 224 | 2011.10 | 140 |
| Zirbitzkogel | 47.072 | 14.560 | 2372 | 2012.10 | 101 |
| Patscherkofel | 47.209 | 11.461 | 2254 | 2013.11 | 337 |
| Feldkirchen | 48.065 | 13.062 | 581 | 2011.10 | 245 |
| Valluga | 47.158 | 10.213 | 2824 | - | 431 |

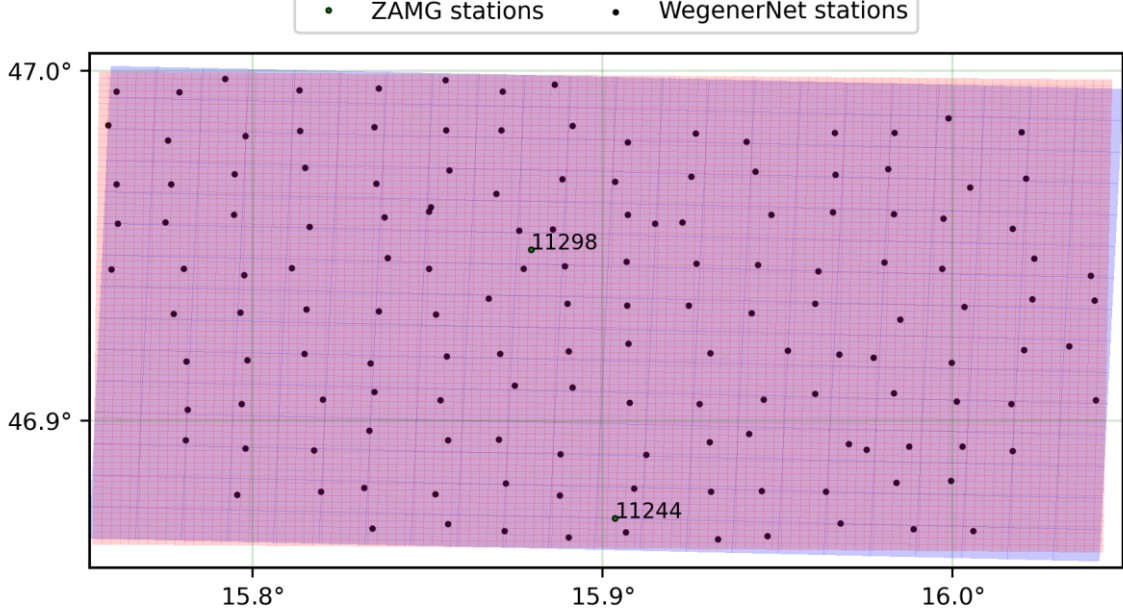


**Figure B1** Locations of WegenerNet stations (black circles) and ZAMG stations (green circles), WegenerNet original (red cells), and after the transforming-regridding process (blue cells).



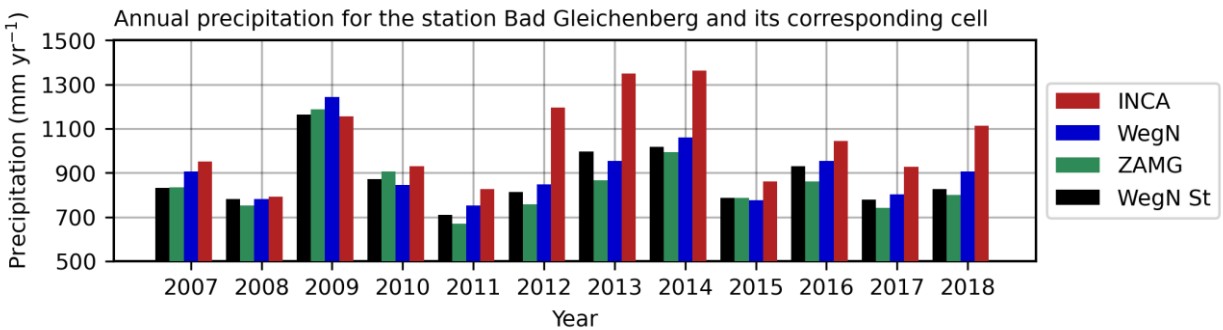

535          **Figure B2** Annual precipitation for the ZAMG station 11244 (Bad Gleichenberg) and its corresponding cell.

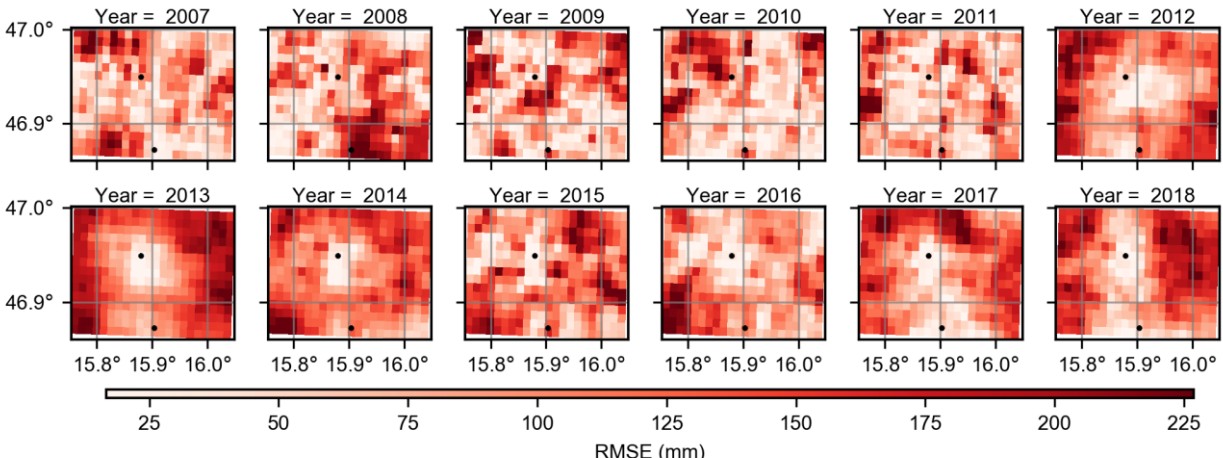

**Figure B3** Annual RMSE in each pixel. The black circles indicate the two ZAMG stations in the study area.



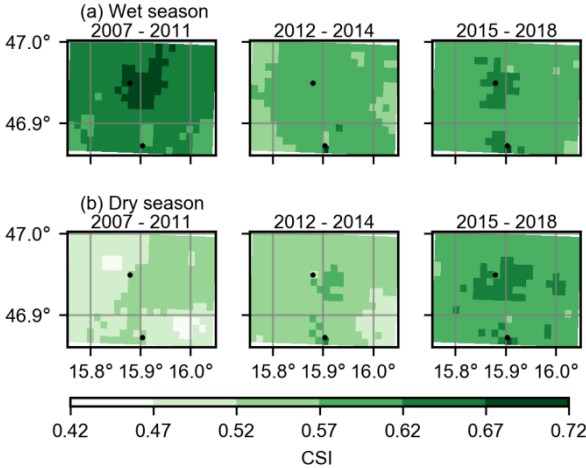

**Figure B4** The CSI values for 2007-2011 (left), 2012-2014 (center), and 2015-2018 (right) in wet (top) and dry (bottom) seasons.
Note that only events with equal or more than 0.1 mm per 15 min were considered. The black circles indicate the two ZAMG stations in
the study area.

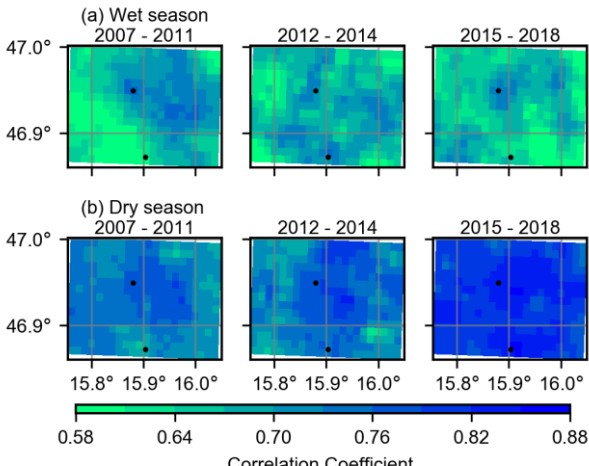

**Figure B5** Correlation coefficient between INCA and WegenerNet for 2007-2011 (left), 2012-2014 (center), and 2015-2018 (right) in wet
(top) and dry (bottom) seasons. The black circles indicate the two ZAMG stations in the study area.





*Author contributions:* EGH and UF set up and designed the study. JF provided in-depth knowledge about WegenerNet data, AK about INCA data. EGH performed the computational and analytical work and prepared the paper, with contributions by UF, JF, and AK. All authors contributed to the interpretation of the results.

*Competing interests:* The authors declare that they have no conflict of interest.

*Acknowledgments:* The authors thank Gottfried Kirchengast and Raoul Collenteur (University of Graz) for fruitful discussions. Furthermore, we would like to thank the Austrian Central Institution for Meteorology and Geodynamics (ZAMG) for the kind provision of INCA data.WegenerNet funding is provided by the Austrian Ministry for Science and
Research, the University of Graz, the state of Styria (which also included European Union regional development funds), and the city of Graz; detailed information can be found online (http://www.wegcenter.at/wegenernet, last access: 16 Jan. 2021).

*Financial Support:* This work is funded by the Austrian Science Fund (FWF) under research grant W1256 (Doctoral Programme Climate Change – Uncertainties, Thresholds and Coping Strategies).


*Data availability:* WegenerNet data products are available at https://wegenernet.org/portal/ (last access: 16 January 2021). INCA data were provided by the Central Institute for Meteorology and Geodynamics (ZAMG) and are not publicly available.





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
