# Peer review of "Evaluation of INCA precipitation analysis using a dense rain gauge network in southeast Austria"

_Hydrology and Earth System Sciences, 2021_

## Author Comment (AC1)

We would like to thank Anonymous referee #1 for his/her positive and insightful comments on the manuscript. We have carefully considered and addressed all of the comments in the following. The original reviewers' comments are in italics, while our response is in plain text.

*The aim of the manuscript is to evaluate the INCA precipitation product. Therefore, the reader can use this study to decide whether INCA is an effective product of high-resolution precipitation for their needs. The authors, though, never clearly arrive at a conclusion about this (except for the last sentence: "Careful consideration must be taken when using merged rain-gauge–radar products, especially in extreme events"). Maybe there should be a discussion section or at least a paragraph where the authors can comment on the evaluation of the INCA product in comparison to other products, and, if possible, radar-rain-gauge blended ones. What have other studies presented in terms of rainfall products performance in Austria or other regions with similar topographic/climatic characteristics? What do these results mean for the INCA performance? Is the model reliable, or other approaches should be preferred?*

As described in the manuscript, there are other studies in this area evaluating precipitation products. However, these studies have been done for shorter time periods (mostly using data before 2015) and different spatial and temporal resolutions. So, a direct quantitative comparison with those studies may not be reliable.
A paragraph will be added at the end of the results section to mention two studies and the INCA performance as below.

Compared to the rapid-INCA product (Kann et al., 2015), the INCA analysis product performs similarly in 2011. For the IMERG products (O et al., 2017), these products underestimate heavy precipitation during 2014-2015, while INCA generally overestimates extremes. One should note that these results may not be entirely comparable since the temporal and spatial resolution of IMERG products are coarser than the INCA analysis product. In general, the INCA analysis product is reliable for hydrological purposes, considering it is a real-time operational product with high temporal and spatial resolutions. Also, the results show that an improvement in the INCA analysis product is taking place. However, the INCA analysis algorithm can be further improved, especially in extreme convective events.

*The authors should also clearly emphasize the contribution of their study. Compared to previous evaluation studies conducted for INCA (Haiden et al., 2011; Kann et al., 2015; Kann and Haiden, 2011), what is the contribution of the present study?*

We agree that more explanation is necessary for the contribution of this study. The following explanation will be added in the revised manuscript.
Haiden et al. (2011) presented the INCA analysis and nowcasting products. Their study verified these products for a summer month and a winter month in 2009 and 2010, respectively. Kann and Haiden (2011) assessed the INCA analysis product for four events in 2008 in four different

regions. Kann et al. (2015) evaluated the rapid-INCA products for the convective season of 2011 (April to September 2011) and four different events in 2011. All those studies are based on INCA data in short time periods before 2012.

However, the purpose of our study is to evaluate the INCA analysis product for 12 years (from 2007 to 2018) to show the changes in the INCA performance due to the new radar installations, improvements in the algorithm, and the addition of more stations to the INCA algorithm. The performance of INCA to estimate precipitation extremes is shown using 12 years of data. In addition, an event-based approach is implemented to extract all the individual events during these 12 years.

*The abstract also does not include the main output of the manuscript, which is whether the INCA precipitation product is eventually a viable choice for hydrological models and decision-making in agriculture and economy (as stated in the beginning). The abstract is also a bit wordy, I think it can be written more concisely.*

Thank you for your suggestion. Based on your comment, we will rewrite the abstract for the revised manuscript.

*Lines 485-486: "We conclude that this overestimation is a result of systematic errors from newly installed radars"*
*Also, in lines 279-282: "From 2012 to 2014, INCA considerably overestimated precipitation in almost all grid cells, and the annual area-mean difference rose to almost 29 % in 2013... We interpret this as an error, introduced by the new radar, which was partly removed by the calibration with ZAMG station data."*
*Maybe I am missing soothing here but it is not clear in the manuscript whether this is the reason for overestimation. To be more specific, in Table B1 we can see that the radars were replaced on 10/2011, 10/2012, and 11/2013. If this is the reason for overestimation for the second period (2012-2014), why does overestimation also occur during the period 2007-2011? (I assume you have considered hydrological years (September to October)):*

We agree with this comment. These sentences should be rewritten to "We conclude that the increase in the overestimation is a result of systematic errors from newly installed radars" and "We conclude that the increase in overestimation in the second period is a result of systematic errors from newly installed radars."

*Line 486: "This overestimation was partly removed in the INCA algorithm using reference gauges."*
*Do you mean that there was no overestimation in the cells near the reference gauge? Maybe a clearer sentence should replace this one.*

The overestimation is lower in the cells near the reference gauges. Since the weight of the gauge estimation decreases with increasing distance from the gauge in the INCA algorithm, we concluded that this overestimation is mostly due to radar estimates and can be partly removed closer to the gauges. We agree that our formulation was a bit unclear and we will change the sentence in the revision.

*Lines 503-505: "In general, INCA has been improving in detecting and estimating precipitation. However, there are errors due to radar estimates and the algorithm for merging radar and rain gauges, which can negatively affect the INCA analysis product."*
*Maybe it is better the conclusions to be understood without the need to read the whole manuscript, you could explain, how is INCA being improved and which are the specific errors.*

We agree with this comment, and we will rewrite this in the revised manuscript.

*Some references are needed in certain sections. Specifically, lines 36-44, 193-195.*

We agree with this comment, and we will add references in the revised manuscript.

*Consider removing the word "very" from the manuscript title. It seems redundant.*

We agree with this comment, and we will correct this in the final manuscript.

*Minor suggestions:*
*• Lines 82-84: Can you give more details about the stations, e.g., average altitude, and also give details about the types of stations presented in Figure 1 in the text. They are not mentioned in the manuscript.*

We agree that there should be more explanation regarding the stations. We will add more explanations in the revised manuscript.

*• Table 1: Consider adding a column with the values which indicate a satisfactory accuracy for each metric*

We will add more explanations about the accuracies of these metrics.

*• Avoid creating one-sentence paragraphs throughout the manuscript*

We have carefully checked this and there will be no more one-sentence paragraph in the revised manuscript.

*• Line 351: change to: "exceeding the 99th quantile"*

We agree with this comment, and we will correct this in the final manuscript.

*• Lines 429 and 439: It is a bit informal to start the paragraphs in this way*

We agree with this comment, and we will correct it in the final manuscript.

*• Lines 490-491: Consider changing the sentence to: "This could be because the INCA algorithm removes false precipitation events and unintentionally..."*

We agree with this comment and we will correct this in the final manuscript.

*References:*

*Haiden, T., Kann, A., Wittmann, C., Pistotnik, G., Bica, B., Gruber, C., 2011. The Integrated Nowcasting through Comprehensive Analysis (INCA) System and Its Validation over the Eastern Alpine Region. Weather and Forecasting 26, 166–183. https://doi.org/10.1175/2010WAF2222451.1*

*Kann, A., Haiden, T., 2011. INCA – an operational nowcasting system for hydrology and other applications. Weather and Forecasting 26, 10.*

*Kann, A., Meirold-Mautner, I., Schmid, F., Kirchengast, G., Fuchsberger, J., Meyer, V., Tüchler, L., Bica, B., 2015. Evaluation of high-resolution precipitation analyses using a dense station network. Hydrol. Earth Syst. Sci. 19, 1547–1559. https://doi.org/10.5194/hess-19-1547-2015*

---

## Author Comment (AC2)

We would like to appreciate Anonymous referee #2 for his/her positive and constructive comments on the manuscript. We have carefully considered and addressed all of the comments in the following. The original reviewers' comments are in italics, while our response is in plain text.

*The main comment relates to the WegenerNet's level 2 data used as reference. This dataset seems to be an interpolation simply based on inverse distance weighting, while the INCA dataset takes into account an external trend caused by topography. I imagine topography plays an important role for the spatial distribution of the rainfall in Austria, why is this not accounted fo in the WegenerNet's level 2 data? This is an important point that should be clarified in the manuscript. How does it impact your comparison and the results? Could the INCA dataset be potentially more accurate because of its account of local topography? This needs to be discussed in the revision because this may have a major impact on the findings.*

> The INCA analysis dataset is generated for the whole of Austria. Since parts of Austria are covered by the Alps and the effect of topography is significant in these parts, this effect is considered in the INCA algorithm. However, the Feldbach region is located in a moderate hilly landscape, and the difference between the highest and lowest altitudes is approximately 300 m. So, we do not expect that topography can have a significant effect on the results in this area. Also, we did not find any systematic effect due to topography in our study.

*Another comment relates to the relatively small area used for analysis of the INCA. How does this apply to the whole of Austria, or even the rest of southeast Austria? I imagine that the INCA dataset is more precise in some areas than in others. For example, the study area is relatively far from the closest radar station used in the radar-gauge merging procedure of INCA. How many rain gauges have been used for the radar-gauge merging of INCA? Where are these gauges located? Is the study area a particularly well- or poorly-covered area, relative to the rest of Austria? All in all, my comment relates to the possibility of extending the results found in this study to the remaining of the INCA dataset for Austria. Do the results of this study apply only to southeast Austria?*

> We agree that there should be more explanation about ZAMG stations and their spatial distribution. We will add more explanations about the stations' density. In general, the average horizontal distance between stations is 18 km.
> Regarding extending these results, there are three reasons that we cannot generalize these results for the whole of Austria: the topography of this area, the climatology of this area, and the distance from the radars. This is also a valid point that the study area is relatively far from the radars, which can negatively affect the INCA performance. However, there is no other network to check this effect separately. Generally, we expect that these results can be representative for other areas with the same topography and climatology (moderate hilly and convective-dominant in summer).

*Minor comments:*
*46: consider rephrasing to "a spatially dense…".*

      We agree with this comment, and we will correct this in the revised manuscript.

*65-66: I found this sentence confusing: "using gridded precipitation fields from the dense WegenerNet weather and climate station" given that the WegenerNet data is not a grid but a set of point measurements with a nearly perfect spatial coverage over the area. This sentence, to me, reads as if you compare an interpolated field with another interpolated field. In Section 2.1 it is made clear that WegenerNet is not an interpolated field, so it would be wise to avoid this confusion in the Introduction of the manuscript. Sentence at L. 93 is also confusing for the same reason. AHA I now understand from L. 97 that this is indeed a gridded dataset! Might be good to reformulate the previous text on it, to let the reader know that you use the gridded data of the WegenerNet dataset.*

      We agree with this comment, and we will rewrite this part in the revised manuscript.

*166-167: aggregated with the sum or mean?*

      We aggregated WegenerNet precipitation to 15-minutes with the sum function. We will add more explanation in the revised manuscript to make this clear.

*224: model performance?*

      Yes, it is model performance. We will correct this in the final manuscript.

*Table 1: as a side note, these indices can be summarized into a single diagram called the target diagram. It would have been useful to have such visualization. Note that this is just a comment, but I do not ask the authors to do it for this manuscript.*

      We appreciate this comment. We will use this comment for future publications.

---

## Author Response (AR1)

**Point-by-point reply, manuscript hess-2021-34**

We would like to thank the editor and reviewers for their constructive comments with regard to our manuscript. In the following, we answer the corresponding comments. The original reviewer comments are in italics, while our response is in plain text.

*The aim of the manuscript is to evaluate the INCA precipitation product. Therefore, the reader can use this study to decide whether INCA is an effective product of high-resolution precipitation for their needs. The authors, though, never clearly arrive at a conclusion about this (except for the last sentence: "Careful consideration must be taken when using merged rain-gauge–radar products, especially in extreme events"). Maybe there should be a discussion section or at least a paragraph where the authors can comment on the evaluation of the INCA product in comparison to other products, and, if possible, radar-rain-gauge blended ones. What have other studies presented in terms of rainfall products performance in Austria or other regions with similar topographic/climatic characteristics? What do these results mean for the INCA performance? Is the model reliable, or other approaches should be preferred?*

As described in the manuscript, there are other studies in this area evaluating precipitation products. However, these studies have been done in shorter time periods (mostly use data before 2015) and different spatial and temporal resolutions. So, a direct quantitative comparison with those studies may not be reliable. A paragraph will be added at the end of the results section to mention these studies briefly as below.

Compared to the rapid-INCA product (Kann et al., 2015), the INCA analysis product performs similarly in 2011, i.e., underestimates precipitation in most cells in the wet season. The IMERG products (O et al., 2017) underestimate heavy precipitation during 2014-2015 in the study area. One should note that this comparison may not be entirely reliable since the temporal and spatial resolutions of IMERG products are coarser than INCA analysis products. In general, the INCA analysis product can be used for different hydrological purposes, considering it is a real-time operational product with high temporal and spatial resolution. Also, the results show that improvements in the INCA analysis product are taking place. It should be noted that INCA precipitation products have a high spatial and temporal resolution, and some errors such as wind drift become more pronounced in higher resolutions. Also, the height that radars can detect precipitation increases with the range from the radar site, which can significantly impact the accuracy of the radar estimates (Harrison et al., 2009). Since the closest radar sees precipitation only above 2000 m from the ground, this can be the main source of uncertainty in detection and estimation of precipitation in the INCA analysis products over the study area. Additionally, there are some sources of uncertainties in the WegenerNet products, such as unheated sensors, wind effects, and the interpolation of data that may have negative effects on the quality of the WegenerNet gridded dataset.

*The authors should also clearly emphasize the contribution of their study. Compared to previous evaluation studies conducted for INCA (Haiden et al., 2011; Kann et al., 2015; Kann and Haiden, 2011), what is the contribution of the present study?*

Thank you for your comments. The following explanation will be added to the final revision. Haiden et al. (2011) presented INCA analysis and nowcasting products. In the study, they verified these products for the whole of Austria for a summer month and a winter month in 2009 and 2010, respectively. Kann and Haiden (2011) assessed the INCA analysis product for four events in 2008 in four different regions. Kann et al. (2015) used WegenerNet station data to evaluate 5 min INCA analysis data (rapid-INCA) for wet season (April-September) of 2011 and 4 different heavy precipitation events. The study showed a general underestimation in rapid-INCA during the wet season. The rapid-INCA also underestimated the average precipitation rate in three out of 4 events. They also showed the roles of rain gauges and radars in rapid-INCA analysis performance.
However, the purpose of this study is to evaluate the INCA analysis product for 12 years (from 2007 to 2018) to show the changes in the INCA performance due to the installation of new radars and improving the INCA algorithm. Also, the performance of INCA to estimate precipitation extremes is shown using 12 years of data. In addition, an event-based approach is implemented to analyze all the individual events during these 12 years.

*The abstract also does not include the main output of the manuscript, which is whether the INCA precipitation product is eventually a viable choice for hydrological models and decision-making in agriculture and economy (as stated in the beginning). The abstract is also a bit wordy, I think it can be written more concisely.*

Thank you for your comment. Based on your comment, we will rewrite the abstract for the final revision.

*Lines 485-486: "We conclude that this overestimation is a result of systematic errors from newly installed radars"*
*Also, in lines 279-282: "From 2012 to 2014, INCA considerably overestimated precipitation in almost all grid cells, and the annual area-mean difference rose to almost 29 % in 2013... We interpret this as an error, introduced by the new radar, which was partly removed by the calibration with ZAMG station data."*
*Maybe I am missing soothing here but it is not clear in the manuscript whether this is the reason for overestimation. To be more specific, in Table B1 we can see that the radars were replaced on 10/2011, 10/2012, and 11/2013. If this is the reason for overestimation for the second period (2012-2014), why does overestimation also occur during the period 2007-2011? (I assume you have considered hydrological years (September to October)):*

We agree with this comment. They should be rewritten to "We conclude that the increase in the overestimation is a result of systematic errors from newly installed radars" and "We conclude that the increase in overestimation in the second period is a result of systematic errors from newly installed radars."

*Line 486: "This overestimation was partly removed in the INCA algorithm using reference gauges."*
*Do you mean that there was no overestimation in the cells near the reference gauge? Maybe a clearer sentence should replace this one.*

We agree with this comment. The overestimation was lower in the cells near the reference gauges. Since the weight of the gauge estimation decreases with increasing distance from the gauge, we concluded that this increase in the overestimation is a result of systematic errors from newly installed radars and can be partly removed closer to the gauges.

*Lines 503-505: "In general, INCA has been improving in detecting and estimating precipitation. However, there are errors due to radar estimates and the algorithm for merging radar and rain gauges, which can negatively affect the INCA analysis product."*
*Maybe it is better the conclusions to be understood without the need to read the whole manuscript, you could explain, how is INCA being improved and which are the specific errors.*

We agree with this comment, and we will rewrite this in the final manuscript.

*Some references are needed in certain sections. Specifically, lines 36-44, 193-195.*

We agree with this comment, and we will add references to the final manuscript.

*Consider removing the word "very" from the manuscript title. It seems redundant.*

We agree with this comment, and we will correct this in the final manuscript.

*Minor suggestions:*
*• Lines 82-84: Can you give more details about the stations, e.g., average altitude, and also give details about the types of stations presented in Figure 1 in the text. They are not mentioned in the manuscript.*

Yes, we agree with this comment, and we will add more explanations in the final manuscript.

*• Table 1: Consider adding a column with the values which indicate a satisfactory accuracy for each metric*

We will add more explanations in this regard in the final manuscript.

*• Avoid creating one-sentence paragraphs throughout the manuscript*

We have corrected this and there will be no more one-sentence paragraph in the next revision.

*• Line 351: change to: "exceeding the 99th quantile"*

We agree with this comment, and we will correct this in the final manuscript.

*• Lines 429 and 439: It is a bit informal to start the paragraphs in this way*

We agree with this comment, and we will correct it in the final manuscript.

*• Lines 490-491: Consider changing the sentence to: "This could be because the INCA algorithm removes false precipitation events and unintentionally..."*

We agree with this comment and we will correct this in the final manuscript.

*References*

*Haiden, T., Kann, A., Wittmann, C., Pistotnik, G., Bica, B., Gruber, C., 2011. The Integrated Nowcasting through Comprehensive Analysis (INCA) System and Its Validation over the Eastern Alpine Region. Weather and Forecasting 26, 166–183. https://doi.org/10.1175/2010WAF2222451.1*

*Kann, A., Haiden, T., 2011. INCA – an operational nowcasting system for hydrology and other applications. Weather and Forecasting 26, 10.*

*Kann, A., Meirold-Mautner, I., Schmid, F., Kirchengast, G., Fuchsberger, J., Meyer, V., Tüchler, L., Bica, B., 2015. Evaluation of high-resolution precipitation analyses using a dense station network. Hydrol. Earth Syst. Sci. 19, 1547–1559. https://doi.org/10.5194/hess-19-1547-2015*

**Referee 2:**

*The main comment relates to the WegenerNet's level 2 data used as reference. This dataset seems to be an interpolation simply based on inverse distance weighting, while the INCA dataset takes into account an external trend caused by topography. I imagine topography plays an important role for the spatial distribution of the rainfall in Austria, why is this not accounted fo in the WegenerNet's level 2 data? This is an important point that should be clarified in the manuscript. How does it impact your comparison and the results? Could the INCA dataset be potentially more accurate because of its account of local topography? This needs to be discussed in the revision because this may have a major impact on the findings.*

Thank you for your suggestion. We will add the following paragraph and further explanations to the final revision.
The WegenerNet and INCA datasets are generated differently regarding topography, i.e., WegenerNet is based on a simple IDW method, while INCA is generated considering the elevation effect. It should be noted that the INCA analysis dataset is produced for the whole of Austria. Since parts of Austria are covered by the Alps and their topography can significantly affect the precipitation estimates, these effects are considered in the INCA algorithm. However,

the Feldbach region is located in a moderately hilly landscape, and the difference between the highest and lowest altitude is approximately 300 m. So, we do not expect that topography can significantly affect the results in this area. Also, we did not find any systematic effect due to topography in our study.

*Another comment relates to the relatively small area used for analysis of the INCA. How does this apply to the whole of Austria, or even the rest of southeast Austria? I imagine that the INCA dataset is more precise in some areas than in others. For example, the study area is relatively far from the closest radar station used in the radar-gauge merging procedure of INCA. How many rain gauges have been used for the radar-gauge merging of INCA? Where are these gauges located? Is the study area a particularly well- or poorly-covered area, relative to the rest of Austria? All in all, my comment relates to the possibility of extending the results found in this study to the remaining of the INCA dataset for Austria. Do the results of this study apply only to southeast Austria?*

We agree that there should be more explanation about ZAMG stations and their distribution. We will add the following explanation about the stations' density.
Note that in general, the average horizontal distance between stations is 18 km.
Regarding extending these results, there are three reasons that we cannot generalize these results for the whole Austria: the topography of this area, the climatology of this area, and the distance from radars. This is also a valid point that the study area is relatively far from radars. This can have a negative effect on INCA performance. Generally, we expect that these results can represent for other areas with the similar topography and climatology (moderately hilly and convective-dominance in summer).

*Minor comments:*
*46: consider rephrasing to "a spatially dense…".*

We agree with this comment, and we will correct this in the final manuscript.

*65-66: I found this sentence confusing: "using gridded precipitation fields from the dense WegenerNet weather and climate station" given that the WegenerNet data is not a grid but a set of point measurements with a nearly perfect spatial coverage over the area. This sentence, to me, reads as if you compare an interpolated field with another interpolated field. In Section 2.1 it is made clear that WegenerNet is not an interpolated field, so it would be wise to avoid this confusion in the Introduction of the manuscript. Sentence at L. 93 is also confusing for the same reason. AHA I now understand from L. 97 that this is indeed a gridded dataset! Might be good to reformulate the previous text on it, to let the reader know that you use the gridded data of the WegenerNet dataset.*

We agree with this comment, and we will correct this in the final manuscript.

*166-167: aggregated with the sum or mean?*

We aggregate to 15-minutes with the sum.

*224: model performance?*

Yes, model performance. We will correct this in the final manuscript.

*Table 1: as a side note, these indices can be summarized into a single diagram called the target diagram. It would have been useful to have such visualization. Note that this is just a comment, but I do not ask the authors to do it for this manuscript.*

We appreciate this comment. We will use this comment for future publications.